# Antimicrobial drug resistant features of *Mycobacterium tuberculosis* associated with treatment failure

Fizza Mushtaq[1,2], Syed Mohsin Raza[1], Adeel Ahmad[3], Hina Aslam[1], Atiqa Adeel[3], Sidrah Saleem[3], Irfan Ahmad[1,2]*

1 Institute of Biomedical and Allied Health Sciences, University of Health Sciences, Lahore, Pakistan,
2 Department of Molecular Biology and Umeå Centre for Microbial Research (UCMR), Umeå University, Umeå, Sweden, 3 Department of Microbiology, University of Health Sciences, Lahore, Pakistan

* irfan.ahmad@umu.se

**Data Availability Statement:** All relevant data are within the paper and its Supporting Information files.

**Funding:** The author(s) received no specific funding for this work.

## Abstract

Tuberculosis stands as a prominent cause of mortality in developing countries. The treatment of tuberculosis involves a complex procedure requiring the administration of a panel of at least four antimicrobial drugs for the duration of six months. The occurrence of treatment failure after the completion of a standard treatment course presents a serious medical problem. The purpose of this study was to evaluate antimicrobial drug resistant features of *Mycobacterium tuberculosis* associated with treatment failure. Additionally, it aimed to evaluate the effectiveness of second line drugs such as amikacin, linezolid, moxifloxacin, and the efflux pump inhibitor verapamil against *M. tuberculosis* isolates associated with treatment failure. We monitored 1200 tuberculosis patients who visited TB centres in Lahore and found that 64 of them were not cured after six months of treatment. Among the *M. tuberculosis* isolates recovered from the sputum of these 64 patients, 46 (71.9%) isolates were simultaneously resistant to rifampicin and isoniazid (MDR), and 30 (46.9%) isolates were resistant to pyrazinamide, Resistance to amikacin was detected in 17 (26,5%) isolates whereas resistance to moxifloxacin and linezolid was detected in 1 (1.5%) and 2 (3.1%) isolates respectively. Among MDR isolates, the additional resistance to pyrazinamide, amikacin, and linezolid was detected in 15(23.4%), 4(2.6%) and 1(1.56%) isolates respectively. One isolate simultaneously resistant to rifampicin, isoniazid, amikacin, pyrazinamide, and linezolid was also identified. In our investigations, the most frequently mutated amino acid in the treatment failure group was Serine 315 in *katG*. Three novel mutations were detected at codons 99, 149 and 154 in *pncA* which were associated with pyrazinamide resistance. The effect of verapamil on the minimum inhibitory concentration of isoniazid and rifampicin was observed in drug susceptible isolates but not in drug resistant isolates. Rifampicin and isoniazid enhanced the transcription of the efflux pump gene r*v1258* in drug susceptible isolates collected from the treatment failure patients. Our findings emphasize a high prevalence of MDR isolates linked primarily to drug exposure. Moreover, the use of amikacin as a second line drug may not be the most suitable choice in such cases.

**Competing interests:** The authors have declared that no competing interests exist.

## Introduction

Tuberculosis is one of the most challenging bacterial infections to treat. It is caused by a group of closely related organisms collectively referred to as the *Mycobacterium tuberculosis* complex (MTC). The MTC includes *M. tuberculosis*, *M. africanum*, *M. orygis*, *M. bovis and the Bacillus Calmette–Guérin strain*, *M. microti*, *M. canetti*, *M. caprae*, *M. pinnipedii*, *M. suricattae and M. mungi* [1]. More than 7 million people are infected with *M. tuberculosis* annually. Tuberculosis was the leading cause of mortality due to an infectious agent before the emergence of Covid-19 pandemic claiming the lives of 1.4 million people in 2019 (Global tuberculosis report, 2020). Pakistan ranks fifth among 22 high burden countries and fourth with respect to multi drug resistant cases of *M. tuberculosis*. Approximately 420,000 cases occur each year in Pakistan and out of these 9000 cases are of drug-resistant TB [2, 3].

Treating *M. tuberculosis* infection has always been very complicated. There is no single drug recommended for TB treatment. Instead, a combination of several drugs must be administrated simultaneously. The panel of 1st line drugs includes pyrazinamide (PZA), ethambutol, rifampicin (RIF) and isoniazid (INH), used in combination therapy for treating non MDR TB patients [4].

Rifampicin resistance primarily develops due to mutations in genes that encode the β-sub-unit of the RNA polymerase (rpoB). Mutations in the *rpoB* gene at codons 531, 516, 511, 526 and 533 are commonly associated with rifampicin resistance, although, rare mutations in other codons of *rpoB* have also been observed [5–7]. These mutations alter the structure of RNA polymerase, preventing rifampicin from binding. Isoniazid (INH) is a prodrug that requires activation by the bacterial enzyme katG. It acts by inhibiting the synthesis of mycolic acid (encoded by the *inhA* gene) which is necessary for mycobacterial cell wall biosynthesis. Mutations in *katG* or *inhA* are primarily associated with isoniazid resistance. The most preva-lent mutations associated with isoniazid resistance are Ser 315 Thr in *katG*, Cys 15 Thr and Thr 8 Cys in *inhA* [8].

Pyrazinamide is also a pro drug. Upon administration, the enzyme pyrazinamidase (*pncA*) converts pyrazinamide to its active form, pyrazinoic acid. Pyrazinoic acid disrupts membrane energetics and inhibits membrane transportation function. Pyrazinamide resistance is mainly associated with mutations in the *pncA* gene [9].

For the treatment of multi-drug resistant TB, injectable drugs such as amikacin, capreomy-cin and kanamycin have been used as second line anti TB drugs. These aminoglycoside drugs inhibit bacterial protein synthesis by blocking the protein synthesis machinery. Point muta-tions in the *rrs* and *eis* genes or in the promoter of the *whiB 7* gene are the most common causes of amikacin resistance [10]. MDR TB additionally resistant to one of the second line drugs and fluoroquinolone are classified as extensivly drug resistant TB (XDR-TB) [11]. To treat such cases, ethionamide, cycloserine, bedaquiline, linezolid and clofazimine are approved as anti-TB drugs [4].

Activation of efflux pumps also contributes to drug resistance phenotype in *M. tuberculosis* [12]. Efflux pumps can maintain sublethal concentrations of antimicrobial drugs by extruding them from the cell. The sublethal concentration provides the bacteria with an opportunity to adapt to antimicrobial drugs and acquire resistance [13]. There are five super families of efflux pumps found to be associated with drug resistance phenotype in *M. tuberculosis*. Among these five super families, ABC (ATP binding cassette) superfamily and MFS (major facilitator super-family) are mainly responsible for efflux pump mediated drug resistance to most of anti TB drugs. Rv1258c, a type of MFS pump, can extrude several TB drugs such as isoniazid, rifampi-cin, ethambutol, and fluoroquinolones [14, 15].

In this study, we present drug resistance features, their associated genetic mutations, and the involvement of efflux pump Rv1258 in the drug resistance of *M. tuberculosis* clinical

isolates obtained from treatment failure tuberculosis patients. Furthermore, we tested the efficacy of second line TB drugs and the efflux pump inhibitor verapamil against these isolates.

## Materials and methods

### Bacterial strains and growth conditions

The study includes a total of 1200 pulmonary tuberculosis patients undergoing treatment with first-line drugs including rifampicin, isoniazid, pyrazinamide, and ethambutol for six months in TB centres located at Mayo Hospital Lahore, and Gulab Devi Hospital Lahore during 2016–2017. A verbal informed consent was obtained from the participants following the guidelines of the institutional ethical review committee at University of Health Sciences, Lahore under letters No; UHS/Education/126-16/2611 and UHS/REG-18/ERC4176. The documentation of patient's information was waved out by the ethical committee due to non-traceable identification of isolates and associated genotypic and phenotypic features with respect to identity of patients.

The treatment of patients with first line drugs was based on identification of non-MDR tuberculosis using GeneXpert MTB/RIF analysis of sputum as a part of routine diagnostic protocol. The effectiveness of treatment in these patients was assessed after 6 months through ZN staining and GeneXpert MTB/RIF analysis of their sputum samples. The specimens detected positive for *Mycobacterium* were further processed for bacterial culture and drug susceptibility testing. *M. tuberculosis* isolates were cultured from the sputum by incubating samples in LJ medium at 37C°. To ensure patients anonymity in accordance with ethical guidelines, the isolates were randomly labelled and kept unidentifiable with respect to the source specimen. The study flow chart is summarized in S1 Fig. Additionally, sixty-four isolates were collected from the sputum of freshly diagnosed TB patients during the same period. All the isolates used in the study are listed in S2 and S3 Tables in S1 File. The isolates were stored in 10% skimmed milk at– 80°C.

To differentiate the *Mycobacterium* isolates at the species level, a multiplex PCR assay was performed as described previously [16]. Upon growth in MGIT (mycobacteria growth indicator tubes; Becton Dickinson) vials, DNA was isolated using a bacterial DNA isolation kit (Qiagen). Multiplex PCR was performed to amplify *rv0577*, and 16S rRNA gene to verify each isolate as *M. tuberculosis*. The primer sequences used in the multiplex PCR are listed in S1 Table in S1 File. Genomic markers of non-tuberculosis species of *Mycobacterium*, including IS*1311*, DT1, *mkan_rs12360* and *mass_3210* were tested to exclude non-tuberculosis species of *Mycobacterium*. Targeted DNA loci detected in individual isolates are shown in S2 Table in S1 File.

### Drug susceptibility testing (DST)

Drug susceptibility testing was performed using MGIT 960 (Becton Dickinson Diagnostic Systems, Sparks, MD) following the recommended protocol as described previously [17, 18]. For this, bacteria were grown in MGIT 960 vials at 37°C by adding 100 μls of bacterial stock (stored in skimmed milk at -80°C) in BACTEC 960 SIRE supplement. On average, it took four days to appear visible growth in MGIT vials. On the fourth day, drug susceptiblity testing was conducted. BACTEC 960 SIRE Supplement (0.8 ml) was added to each MGIT tube. Aseptically, 0.1 ml of properly reconstituted drug was added into each tube at different concentrations. Aseptically, 0.5 ml of the well-mixed culture suspensions diluted with water (1:100) were added into each of the drug containing tubes except the growth control (GC) tube. For control, test culture suspensions were diluted to 50 times with normal saline. Inoculated MGIT vials were mixed well by gently inverting the tubes several times and loaded into MGIT 960. The

instrument interpreted the susceptibility results when the growth unit (GU) in the growth control reached 400 units (within 4–13 days). At this point, the GU values of the drug vial were evaluated according to the following criteria:

S = Susceptible–the GU of the drug tube is less than 100

R = Resistant–the GU of the drug tube is 100 or more.

Isolates were considered sensitive to the drug upon the growth at the following concentration for the respective drug: rifampicin 1.0 μg/ml, isoniazid 0.1 μg/ml, amikacin, 1.0 μg/ml, moxifloxacin 0.125μg/ml and linezolid 1.0 μg/ml.

### Targeted DNA sequencing

Genomic DNA from *M. tuberculosis* isolates was extracted using a bacterial DNA extraction kit (Qiagen). Using isolated DNA as a template, the loci corresponding to *rpoB*, *katG*, *pncA*, *eis*, *rrs* and the promoter of *whiB 7* gene were amplified using the primers listed in S1 Table in S1 File. The amplified PCR products were purified for DNA sequencing. The DNA sequence of each locus was aligned with the corresponding sequence of *M. tuberculosis H37Rv*.

### *In silico* analysis of novel mutations in PncA

The structural model of the mutated protein was generated using the "mutation" tool of Swiss-PDB Viewer ver. 4.10, which allows browsing through a rotamer library to substitute amino acids [19]. The mutant structure was saved in the ".pdb" format. Both structures were energy minimized by GROMOS96 conjugate gradient method [20]. For superimposition and root-mean-square deviation (RMSD) calculations of the atomic positions in the mutated modeled protein, UCSF Chimera 1.15 was used [21].

### RT–qPCR analysis of putative efflux pumps

All of the tested strains were treated with 0.5 μg/ml of rifampicin or 0.5 μg/ml isoniazid for one hour in MGIT vials. Total RNA was extracted with and without treatment of the drug using a total RNA isolation kit (Thermo scientific). Subsequently, the isolated RNA was treated with DNAase for 1 hour. The integrity of RNA was tested by agarose gel electrophoresis. The relative expression level of the efflux pump gene *Rv1258c* was assessed by RT-qPCR using the oligonucleotides described in S1 Table in S1 File. A first strand DNA synthesis kit (thermo scientific) was used for cDNA Synthesis. Real-time PCR was performed using SYBR Green real time PCR master Mix (Thermo scientific). Relative mRNA levels were determined using the comparative quantification cycle (Cq). The fold change in the mRNA levels of each gene was calculated in the presence and the absence of the drug. Each experiment was conducted in triplicate by collecting RNA from three biological replicates. The mRNA levels were normalized using the *M. tuberculosis* 16S rDNA as an internal control for each experiment and presented as the mean-fold change (±*SD*) compared to the control.

## Results

### Hetero resistance to amikacin and pyrazinamide displayed by MDR *M. tuberculosis* cured from treatment failure patients as revealed from a comparative retro perspective cohort study

In our investigation of the treatment response in 1200 MTB patients, sixty-four patients remained positive for *M. tuberculosis* after six months of treatment with first line drugs as detected through ZN staining of sputum. Among these sixty-four isolates, 46 (71.9%) were

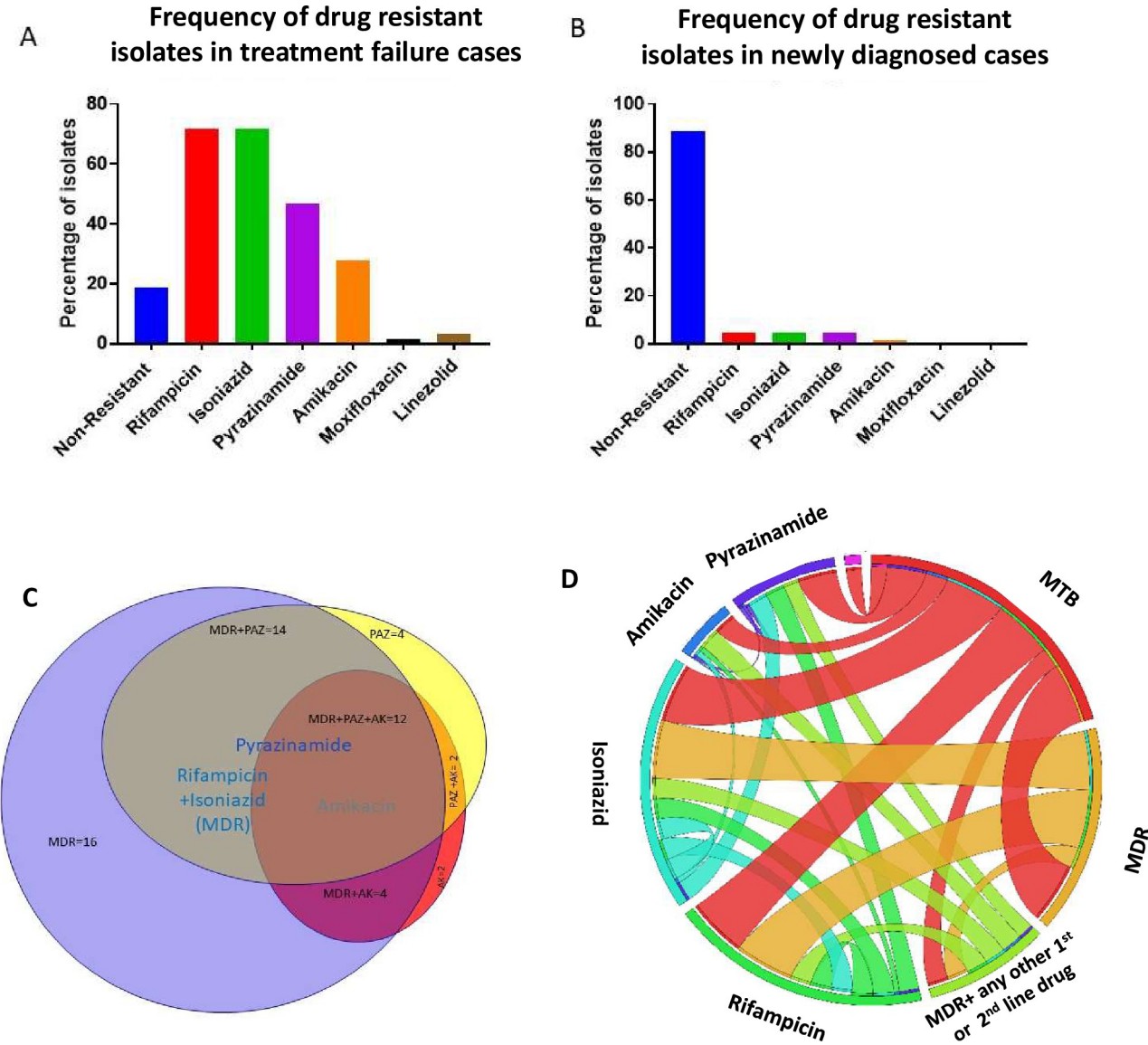

**Fig 1. Frequency of drug resistant isolates in treatment failure cases and treatment naive cases.** Bar chart diagram representing the percentage of drug resistant isolates among treatment failure (A) and treatment naive cases (B). Total number of isolates tested in each group were 64. (C) Venn diagram representing the frequency of mono drug resistance and different combinations of extensive drug resistant isolates. Each drug is represented by a coloured circle weighted by the prevalence of phenotypic resistance to that drug. The circle in purple represents rifampicin and isoniazid resistant isolates, the circle in yellow represents pyrazinamide resistant isolates and the circle in red represent amikacin resistant isolates. The overlapping regions represent different combinations of extensive drug resistant isolates. (D) Circos diagram to express frequency of *M. tuberculosis* (MTB) isolates with respect to different combinations of drug resistance.

resistant to rifampicin and isoniazid. Importantly, all rifampicin resistant isolates were also found to be resistant to isoniazid (Fig 1A, Table 1, S3 and S4 Tables in S1 File).

Resistance to pyrazinamide was observed in 30 (46.9%) isolates. The isolates were further tested for the susceptibility to 2nd line anti MTB drugs amikacin, moxifloxacin and linezolid. Seventeen (26.6%) isolates were found to be resistant to amikacin while one isolate was resistant to moxifloxacin and two isolates were resistant to linezolid. Notably, twelve isolates (18.8%) from the treatment failure cases were susceptible to all tested first and second-line drugs (Fig 1A, Table 1 and S1 Table in S1 File). Among the sixty-four isolates collected from

**Table 1. Frequency of drug resistant *Mycobacterium tuberculosis*.**

| N = 128 | Pan drug sensitive | Pyrazinamide (PAZ) Resistant | Amikacin (AK) Resistant | Rifampicin + Isoniazid (MDR) | MDR + PAZ | MDR +AK | MDR+AK +PAZ | MDR+AK +LZ | MDR+AK +PAZ+ LZ |
|---|---|---|---|---|---|---|---|---|---|
| **Treatment failure Isolates N = 64** | 12(18.75%) | 30 (46.9%) | 17 (26,5%) | 46 (71,9%) | 15 (23,4%) | 4(6,2%) | 12 (18,6%) | 1 (1.56%) | 1 (1.56%) |
| **Treatment naive isolates N = 64** | 57(89,6%) | 3 (4.7%) | 1(1,5%) | 3 (4.7%) | 0 | 0 | 0 | | |
| | | | | 3 (4.7%) | | 0 | | | |

freshly diagnosed TB cases, three (4.6%) isolates were found to be resistant to rifampicin and isoniazid, three (4,6%) isolates were resistant to pyrazinamide and, one (1.6%) isolate was resistant to amikacin (Fig 1B, Table 1 and S5 Table in S1 File). Resistance to moxifloxacin and linezolid was not detected in the treatment naive group.

Among MDR isolates (resistant to rifampicin and isoniazid), fourteen isolates were additionally resistant to pyrazinamide and sixteen isolates were additionally resistant to amikacin (Table 1, Fig 1C and 1D). Alarmingly, twelve isolates (18.8%) were resistant to rifampicin, isoniazid, pyrazinamide and amikacin, the four anti MTB drugs (Fig 1C and 1D, Table 1, S3 and S4 Tables in S1 File). The difference in drug resistance between the drug failure and treatment naive groups of isolates was expected considering that the isolates cured from the treatment failure patients had been exposed to drugs for six months. However, the high frequency of amikacin resistant isolates in the treatment failure group was unexpected, as amikacin was not used to treat these patients. This observation requires further investigation to understand the role of first line drugs in the induction of amikacin resistance. These findings suggest that the use of amikacin as a second line drug is remarkably less appropriate as compared to moxifloxacin and linezolid.

## Mutations associated with multi drug resistance and novel mutations in *pncA* of extensive drug resistant *M. tuberculosis*

Among the rifampicin resistant isolates, five single nucleotide mutations were detected in the *rpoB* gene of 46 isolates from the treatment failure cases (Table 2).

The most frequent mutation was the replacement of serine 531 with leucine followed by the replacement of aspartic acid 516 with valine or tyrosine. The replacement of leucine 511 with arginine was detected in two isolates and rare mutation resulting in the replacement of threonine 508 with isoleucine was found in one isolate. Most of the mutations were in *rpoB* mutational hot spot region known to be a target for rifampicin resistance. Among isoniazid resistant isolates, mutations leading to the replacement of serine 315 with threonine and serine 315 with asparagine in *katG* was detected in twenty-four and four isolates respectively. Notably, serine 315 of *katG* was the most frequently altered amino acid residue in the entire collection of MDR, isolates. DNA sequencing of the *rplC* gene of the linezolid resistant isolate MTBLH021 detected a mutation leading to substitution of methionine 153 with threonine.

Surprisingly, *rrs* and *eis* loci in all amikacin resistant isolates were identical to the corresponding loci of *H37 Rv* strain indicating that amikacin resistance in these isolates is independent of *rrs* and *eis*. Moreover, the mutations in the promoter region of *whiB 7* is also known to cause amikacin resistance. However, DNA sequencing of the promoter region of *whiB 7* in all amikacin resistant isolates did not identify any alteration with respect to *whiB 7* promoter of *H37 Rv* isolate.

Among pyrazinamide resistant isolates, thirteen single nucleotide mutations in the *pncA* gene were detected (Table 2, Fig 2). Notably, the mutations leading to amino acid changes were not confined to a specific codon but varied from isolate to isolate (Table 2). Additionally,

**Table 2. Mutations detected in *M. tuberculosis* isolates from treatment failure cases.**

|  | Targeted gene | Codon Position | Mutation | A.A Change | Mutation Classification | Mutation frequency | Isolate ID |
|---|---|---|---|---|---|---|---|
| Rifampicin | *rpoB* | 508 | ACC→ATC | Thr→Ile | NS | 1 | MTBLH-54 |
|  |  | 511 | CTG→CGG | Leu→Arg | NS | 2 | MTBLH-07, MTBLH-10 |
|  |  | 516 | GAC→TAC | Asp→Tyr | NS | 3 | MTBLH-07, MTBLH-10, MTBLH-52 |
|  |  | 516 | GAC→GTC | Asp→Val | NS | 3 | MTBLH-28, MTBLH-45, MTBLH-50 |
|  |  | 526 | CAC→AAC | His→Asn | NS | 3 | MTBLH-41, MTBLH-42, MTBLH-24 |
|  |  | 531 | TCG→TTG | Ser→Leu | NS | 5 | MTBLH-23, MTBLH-25, MTBLH-26, MTBLH-29, MTBLH-31 |
| Isoniazid | *katG* | 295 | CAG→CAA | Gln→Gln | NS | 1 | MTBLH-63 |
|  |  | 315 | AGC→ACC | Ser→Thr | NS | 24 | MTBLH-01, MTBLH-07, MTBLH-11, MTBLH-25, MTBLH-17, MTBLH-27, MTBLH-28, MTBLH-29, MTBLH-16, MTBLH-37, MTBLH-39, MTBLH-40, MTBLH-41, MTBLH42, MTBLH-44, MTBLH-46, MTBLH-48, MTBLH-50, MTBLH-51, MTBLH-53, MTBLH-54, MTBLH-59, MTBLH-61, MTBLH-62 |
|  |  | 315 | AGC→AAC | Ser→Asn | NS | 4 | MTBLH-23, MTBLH-38, MTBLH-45, MTBLH-55 |
| Amikacin | *Rrs* | ND | ND | ND | NA | ND | ND |
|  | *Eis* | NA | ND | ND | NA | NA | ND |
|  | *Whib7* (Promotor) | NA | NA | NA | NA | NA | ND |
| Pyrazinamide | *pncA* | 7 | GTC→TTC | Val→Phe | NSM | 02 | MTBLH-59, MTBLH-61 |
|  |  | 65 | TCC→TCT | Ser→NC | SM | 08 | MTBLH-25, MTBLH-29, MTBLH-38, MTBLH-43, MTBLH-55, MTBLH-08, MTBLH-57, MTBLH-62 |
|  |  | 76 | ACT→CCT | Thr→Pro | NSM | 02 | MTBLH-57, MTBLH-62 |
|  |  | 94 | TTC→CTC | Phe→Leu | NSM | 01 | MTBLH-18 |
|  |  | 99 | TAC→Del. | Tyr→ | FS (Novel) | 01 | MTBLH-43 |
|  |  | 108 | GGA→CGA | Gly→Arg | NSM | 01 | MTBLH-26 |
|  |  | 119 | TGG→AGG | Trp→Arg | NSM | 01 | MTBLH-18 |
|  |  | 135 | ACC→CCC | Thr→Pro | NSM | 01 | MTBLH-21 |
|  |  | 141 | CAG→CCG | Gln→Pro | NSM | 01 | MTBLH-40 |
|  |  | 149 | AAT→Del. | Asn→ | FS (Novel) | 02 | MTBLH-38, MTBLH-55 |
|  |  | 151 | TTG→TCG | Leu→Ser | NSM | 01 | MTBLH-37 |
|  |  | 154 | AGG→ATG | Arg→Met | NSM (Novel) | 01 | MTBLH-20 |
|  |  | 172 | CTG→Del. | Leu→ | FS (Novel) | 01 | MTBLH-60 |
| Linezolid | *rplC* | 153 | ATG→ACG | Met →Thr | NSM | 01 | MTBLH-21 |

*NS: Non-Synonymous mutation

**S: Synonymous mutation

***FSM: Frameshift Mutation, Del: Deletion, ND: No mutation detected, NA: Not Applicable

three novel frame shift mutations were detected at codon 99, 149 and 172 in isolates MTBLH043, MTBLH038 and MTBLH055 respectively. Furthermore, one novel mutation converting arginine 154 to methionine was found in one isolate. 3-D modelling of protein structure revealed that the arginine 154 to methionine mutant retained two hydrogen bonds with valine 128 and valine 130 but lost three hydrogen bonds present between the wild-type arginine154 and aspartic acid 129. The sidechain of methionine 154 also produced a clash with arginine 02 side chain suggesting a significant destabilization in structure upon these mutations (Fig 2).

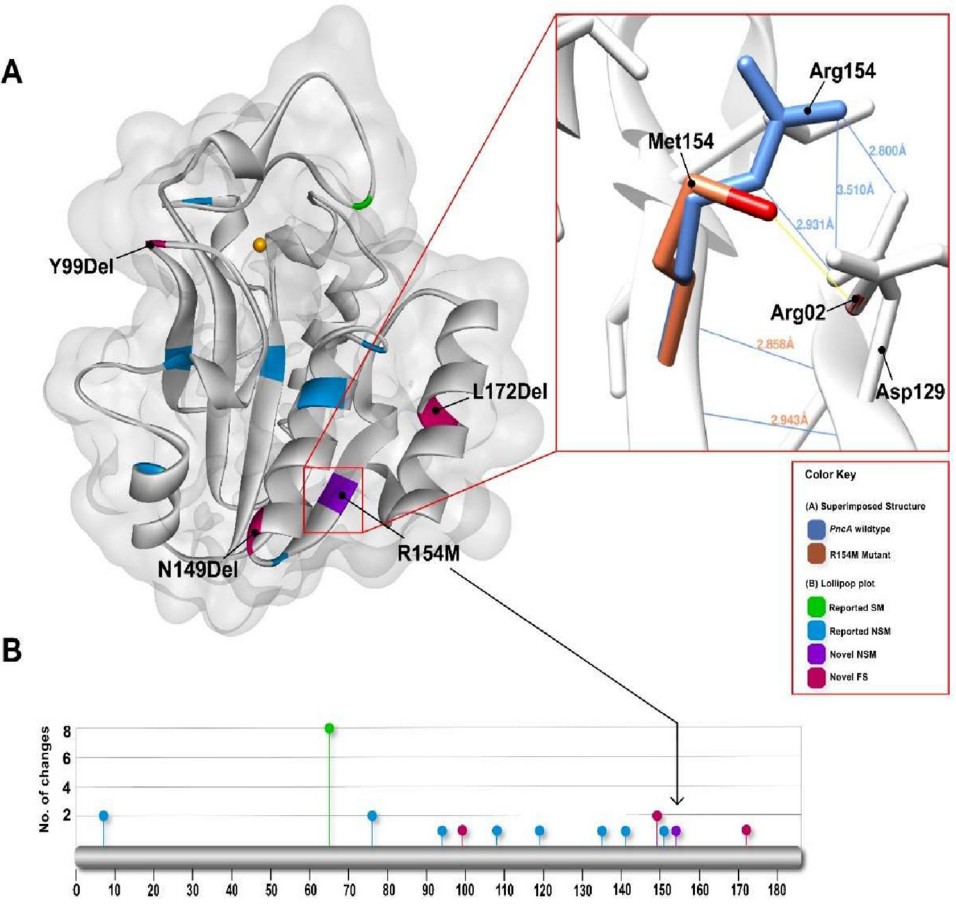

**Fig 2. 3-D model of PncA, highlighting mutations identified in pyrazinamide resistant isolates.** (A) The amino acids detected mutated in pyrazinamide resistant isolates is shown with colour codes. The effect of novel mutation arginine 154 to methionine is shown by superimposing the native protein structure with mutated one (B) the frequency of mutations found in *pncA* is shown with scale bar diagram.

## Interplay of intrinsic and efflux pump mediated drug resistance in *M. tuberculosis*

Among treatment failure cases, there were twelve isolates detected susceptible to all tested TB drugs. Three of these isolates were randomly chosen for further investigation on the expression of the efflux pump associated gene *Rv1258* in the presence and absence of suboptimal concentrations of isoniazid and rifampicin using RT-qPCR analysis. Additionally, three drug susceptible isolates collected from freshly diagnosed TB patients and three MDR isolates resistant to rifampicin and isoniazid were tested for alteration in the expression of efflux pump associated genes upon the exposure to rifampicin and isoniazid. The expression of *rv1258* was significantly increased upon the exposure to rifampicin and isoniazid in the drug susceptible isolates (Fig 3A). In MDR isolates, although there was a trend of increased Rv1258 expression in two isolates, the difference was not statistically significant. In drug susceptible isolates recovered from treatment failure patients, both rifampicin and isoniazid enhanced the expression of Rv1258 2-3-fold, whereas the increase in treatment naive isolates was less than 2-fold (Fig 3A). The difference in Rv1258 expression between the treatment naive and treatment failure groups was statistically significant. This finding led us to speculate that efflux pumps contribute to the tolerance to rifampicin and isoniazid in isolates recovered from patients treated with these drugs.

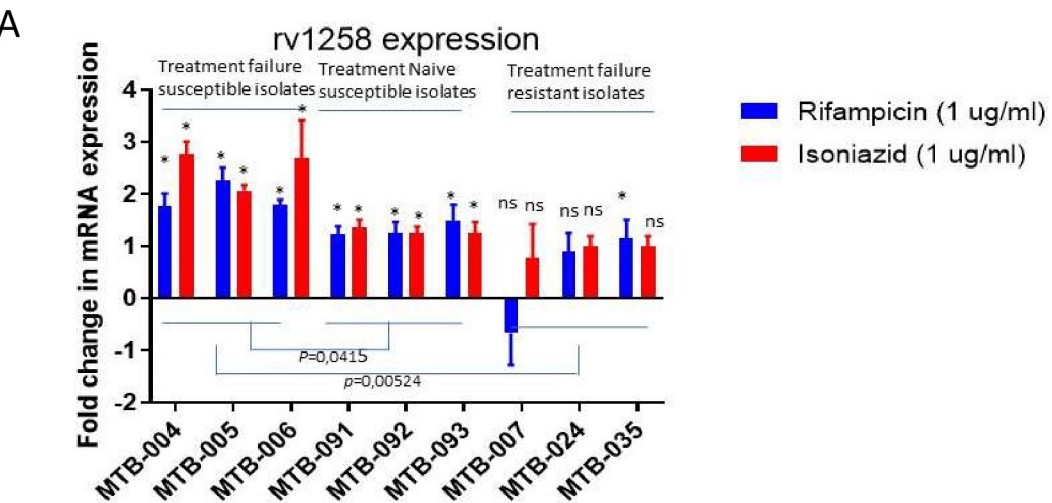

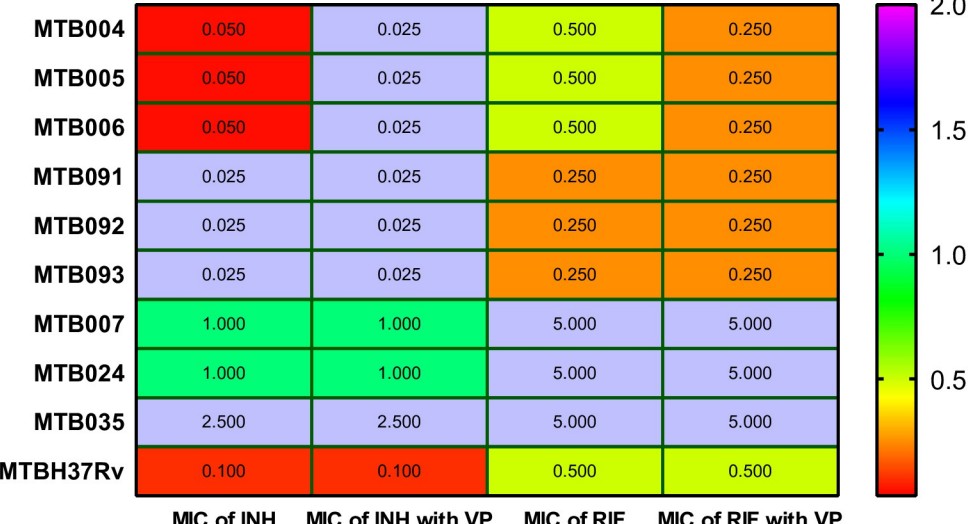

**Fig 3. The effect of rifampicin and isoniazid on the transcription of *rv1258* in *M. tuberculosis* treatment failure isolates in comparison with MDR and treatment naive isolates.** (A) Column bar chart representing the fold change of the mRNA levels of rv1258 upon the treatment with rifampicin or isoniazid. The statistical significance of the difference in mRNA levels with or without treatment of drug was measured by non-parametric *t* test. Significance defined by *p* value <0,05. The difference in the fold change mRNA levels between the groups i.e treatment failure group versus treatment naive group was calculated by two-way ANOVA. Significance defined by *p* value <0,05. (B) Heat map demonstrating the effect of verapamil on minimum inhibitory concentration of rifampicin and isoniazid in treatment failure drug susceptible isolates (MTBLH 04, 05, 06), treatment naïve drug susceptible isolates (MTBLH 91, 92, 93), drug resistant isolates (MTBLH 07, 24, 35) and *M. tuberculosis* H37rV.

## The impact of efflux pump inhibitor verapamil on susceptibility to TB drugs in MDR and pan-drug-susceptible isolates

In order to check if efflux pump inhibitor and membrane stress response inducer verapamil could enhance the drug sensitivity of drug resistant *M. tuberculosis*, the minimum inhibitory concentrations of rifampicin and isoniazid were determined by treating isolates with 64 μg/ml

of verapamil (Fig 3B). The MICs of isoniazid and rifampicin in all tested non-MDR isolates from treatment failure patients were 0.05 µg/ml and 0.5 µg/ml respectively whereas, the MICs of isoniazid and rifampicin in all tested non MDR isolates from treatment naive patients were 0.025 µg/ml and 0.25µg/ml respectively. The treatment of verapamil reduced the MICs of drugs in non-MDR treatment failure isolates to the level of treatment naive non-MDR isolates. Altogether, these findings suggest that efflux pump activation plays a partial role in mediating drug resistance particularly, in isolates that have not acquired intrinsic drug resistance. However, further in vivo studies are required to establish whether this kind of efflux pump mediated drug resistance can lead to treatment failure.

To further investigate the effect of verapamil against drug-resistant *M. tuberculosis*, thirty-three MDR isolates were tested for susceptibility to verapamil in the presence and absence of commonly used TB drugs. In 29 MDR isolates, the MIC of verapamil was 256 µg/ml whereas in four isolates the MIC of verapamil was 512 µg/ml. Subsequently, the impact of verapamil on susceptibility to rifampicin, isoniazid and amikacin was assessed at different concentrations of verapamil. However, verapamil at any concentration below its MIC did not revert resistance to rifampicin, isoniazid, and amikacin in any of the MDR isolates (S6-S8 Tables in S1 File). These findings indicate that verapamil may have a preventive role overcoming adaptive drug resistance rather than reverting intrinsic resistance.

## Discussion

Here, we present a detailed investigation of drug resistant features of *M. tuberculosis* associated with treatment failure. The incidence of treatment failure was observed in 5.33% (64 out of 1200) patients initially infected with non-MDR *M. tuberculosis* strains (Fig 1, Table 1). The course of treatment induced multi drug resistance in 81.5% (52/64) patients whereas the drug resistance was not the main cause of treatment failure in 12/64 patients. Interestingly, resistance to both rifampicin and isoniazid was found to coexist in MDR isolates (Fig 1), despite the different mechanisms of action of these two drugs. However, mutations associated with isoniazid resistance were relatively conserved and mostly located within the codon 315 of *katG* gene, while there was substantial variability in the mutation sites leading to rifampicin and pyrazinamide resistance (Table 2). Furthermore, our study identified several novel mutations in the *pncA* gene associated with pyrazinamide resistance (Table 2). Notably, three of these mutations caused frameshifts in the open reading frames, resulting in modified *pncA* gene products (Fig 2).

Second-line drugs, specifically moxifloxacin and linezolid, demonstrated high efficacy against the treatment failure MDR isolates, aligning with findings from other regions around the world [22]. However, the unexpectedly high prevalence of amikacin-resistant isolates (26.5%) is a cause for concern, as it further restricts the already limited options for anti-TB drugs. Previous reports typically indicated amikacin resistance in *M. tuberculosis* to be below 10% [22–24]. Intriguingly, genetic mutations associated with amikacin resistance in the *rrs* and *eis* genes, as well as the *whiB* 7 promoter, were not identified in these amikacin-resistant isolates. Given that these patients were not receiving amikacin treatment, it raises questions about whether exposure to other antibiotics within the first-line drug panel contributes to amikacin resistance. Conducting systematic phenotypic and genetic analyses of *M. tuberculosis* treated with these drugs may reveal further insights.

It's worth noting that multi-drug resistance was not the primary cause of treatment failure in 12 out of the 64 patients, as these isolates remained pan-drug susceptible (Table 1). Although the accurate cause of treatment failure in such non-MDR isolates remains to investigate. The higher expression of efflux pump gene *rv1258c* in these isolates suggests a potential

contribution of efflux pump activation to treatment failure (Fig 3A). The activation of Rv1258C efflux pump in response to rifampicin treatment is well evident in even in macrophages environment [25, 26]. Other modes of adaptive drug resistance [27], two component regulatory systems [28], alterations in membrane envelope lipoproteins [29] and biofilm formation [30] may also have contribution in the treatment failure. However, clinical data linking these modes of resistance to treatment failure remains limited. In the complex and variable in vivo, environment certain conditions may activate two component systems, efflux pumps or other regulatory responses, ultimately preventing drugs from reaching their targets. Although, resistance mediated by efflux pumps or other regulatory responses relevant during infection is not reflected in routine invitro drug susceptibility testing, the functionality of these pumps against antimicrobial drugs in the macrophage environment and in vivo, mouse infection models is well documented [31–33].

Verapamil is recognized as an effective efflux pump inhibitor in *M. tuberculosis*, reducing the minimum inhibitory concentration of several TB drugs [34–36]. However, it should be noted that verapamil doesn't directly inhibit efflux pumps but exerts anti-mycobacterial activity by affecting membrane potential [37]. Surprisingly, we did not observe an effect of verapamil on the MICs of rifampicin, isoniazid, and amikacin in treatment failure MDR isolates. Nevertheless, verapamil did reduce the MICs of isoniazid and rifampicin in non-MDR isolates (Fig 3B). These findings provide a clue that adaptive drug resistance associated with efflux pump activation may be treated using verapamil as an adjuvant therapy.

In summary, our study aimed to analyze the molecular biology of *M. tuberculosis* linked to treatment failure. The results underscore the importance of more frequent phenotypic and genotypic susceptibility testing throughout the course of treatment. However, a notable limitation of our study is the lack of detailed patient information, including dietary habits, co-morbidities, and medications. Obtaining such information could significantly enhance our understanding of the factors contributing to drug resistance during treatment. Hence our findings trigger the need of an extensive surveillance study involving a larger patient population, with access to comprehensive clinical data, to assess the impact of patient lifestyle, immune status, and environmental factors on the development of drug resistance associated with treatment failure.

## Supporting information

**S1 Fig.**
(TIF)

**S1 File.**
(DOCX)

## Acknowledgments

We are grateful to the TB centres at Mayo Hospital Lahore and Gulab Devi hospital Lahore and Unique Lahore Laboratory for giving access to clinical isolates.

## Author Contributions

**Conceptualization:** Syed Mohsin Raza, Adeel Ahmad, Sidrah Saleem, Irfan Ahmad.

**Data curation:** Fizza Mushtaq, Syed Mohsin Raza, Adeel Ahmad, Hina Aslam, Atiqa Adeel, Irfan Ahmad.

**Formal analysis:** Fizza Mushtaq, Syed Mohsin Raza, Adeel Ahmad, Hina Aslam, Irfan Ahmad.

**Investigation:** Fizza Mushtaq, Hina Aslam, Atiqa Adeel.

**Methodology:** Fizza Mushtaq, Syed Mohsin Raza.

**Resources:** Sidrah Saleem, Irfan Ahmad.

**Software:** Syed Mohsin Raza.

**Supervision:** Irfan Ahmad.

**Validation:** Fizza Mushtaq.

**Visualization:** Atiqa Adeel.

**Writing – original draft:** Irfan Ahmad.

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
