## [Decision Letter · Decision Letter 0]

15 Jun 2023

PONE-D-22-27870Antimicrobial drug resistant features of Mycobacterium tuberculosis associated with treatment failure in PakistanPLOS ONE

Dear Dr. Ahmad,

Thank you for submitting your manuscript to PLOS ONE. After careful consideration, we feel that it has merit but does not fully meet PLOS ONE’s publication criteria as it currently stands. Therefore, we invite you to submit a revised version of the manuscript that addresses the points raised during the review process. Please submit your revised manuscript by 1 July, 2023. If you will need more time than this to complete your revisions, please reply to this message or contact the journal office at plosone@plos.org. Please include the following items when submitting your revised manuscript:A rebuttal letter that responds to each point raised by the academic editor and reviewer(s). You should upload this letter as a separate file labeled 'Response to Reviewers'.A marked-up copy of your manuscript that highlights changes made to the original version. You should upload this as a separate file labeled 'Revised Manuscript with Track Changes'.An unmarked version of your revised paper without tracked changes. You should upload this as a separate file labeled 'Manuscript'.

We look forward to receiving your revised manuscript.

Kind regards,

Atul Vashist, PhD

Academic Editor

PLOS ONE

3. Thank you for stating the following in the Acknowledgments/ Funding Section of your manuscript:

“IA received support from Swedish Research council (2020-06136) and Higher Education Commission Pakistan (8666/Punjab/NRPU/R&D/HEC/2017). FM, AA, SK and HA received support from University of Health Sciences, Lahore”

“The author(s) received no specific funding for this work”

Reviewers' comments:

Reviewer's Responses to Questions

**Comments to the Author**

1. Is the manuscript technically sound, and do the data support the conclusions?

Reviewer #1: Yes

Reviewer #2: Partly

2. Has the statistical analysis been performed appropriately and rigorously? 

Reviewer #1: I Don't Know

Reviewer #2: No

3. Have the authors made all data underlying the findings in their manuscript fully available?

Reviewer #1: Yes

Reviewer #2: Yes

4. Is the manuscript presented in an intelligible fashion and written in standard English?

Reviewer #1: Yes

Reviewer #2: No

5. Review Comments to the Author

Reviewer #1: This study highlights the drug resistance profiles and related gene mutations in M. tuberculosis in Pakistan based on phenotypic and genotypic studies. As M. tuberculosis drug resistance including MDR and XDR phenotypes are a serious concern worldwide, it is of great importance to look at the antibiotic resistance patterns for proper treatment and patient management. Therefore this report is a good compilation of work. However, I have some concerns as mentioned below, and upon clarifying some issues, the manuscript can be accepted for publication in PloS One.

Abstract:

-Please change the starting phrase in abstract to a correct way: "To treat tuberculosis is relatively a complicated procedure……

- in the abstract results section, please add the number and % of MDR and XDR cases.

Introduction:

-According to the reference 4, the first line anti-TB drugs for the treatment of non-MDR cases of TB (group 1) are Isoniazid, rifampicin, pyrazinamide and ethambutol as core drugs, streptomycin is no longer used routinely.

So why you included the latter in the anti-TB regimen? How you manage to use that as an injectable drug for duration of treatment? Please explain that in the discussion.

Methods:

-The information regarding the cases should well-explain at the beginning of methods section, for instance 1200 patients were confirmed TB cases, among them you found 64 drug-resistant cases and for comparison you used the same number naïve cases.

-What does (65-128) means?

-In Drug susceptibility testing (DST) section, please put the reference number together as [16-18]. However why you included 3 references for a single method?

Results:

-As I noticed the rate of MDR was high in this study, again is better when you are talking about resistance to Isoniazid and Rifampicin, add the word "simultaneous" to emphasis that they are MDR. I suggest to re-write the sentence as:

"Among these sixty-four isolates, 46 isolates (71.9%) were simultaneously resistant to rifampicin and isoniazid (MDR phenotype)."

-In naïve group you had also 4.6% MDR (with simultaneous resistance to rifampicin and isoniazid ), you did not explain in the discussion about these, since it seems that these are the cases of primary resistance. Have you investigated the primary and secondary resistance in this research project? This should be included as the information is very useful about the circulating resistant phenotypes in a community.

-you have bold the amikacin resistance in your study when is used for the treatment of MDR –TB cases. According to WHO the drug of choice for the treatment of first-line anti-TB treatment failure are aminoglycosides and fluoroquinolones. I did not find any statements in this section regarding the number and rate of XDR-TB, however in Table 2 it is mentioned.

What was your criteria for determining XDR strain? Since according to the WHO definition for XDR (resistance to fluoroquinolones and one of the injectable aminoglycosides), and you reported only one case with moxifloxacin.

-For XDR cases, please add a column in Table 1 and explain more in discussion section.

Reviewer #2: The authors in this study entitled “Antimicrobial drug resistant features of Mycobacterium tuberculosis associated with treatment failure in Pakistan” try to elucidate antimicrobial drug resistant features of Mycobacterium tuberculosis associated with treatment failure and offer appropriateness of the erstwhile second line drugs and the efflux pump inhibitor verapamil against M. tuberculosis isolates associated with treatment failure. The manuscript targets an important issue; however, it needs to be substantially improved before consideration for publication. The authors should also perform a general proof reading for typographical errors in the manuscript. My comments are as below.

• WHO has recently changed the algorithm of Drug resistant TB treatment and redefined XDR-TB. Please discuss results according the new treatment guidelines. Even though the patients were collected in 2 sets one in 2016-17 and other recently. When were fresh samples collected? How were these patients treated? How long was the follow up done?

• What was the status of DST for these patients when diagnosed, when they were put on 1st line treatment. What were the results of the molecular tests for these patients, was MGIT-DST, Xpert or LPA done on these patients?

• The authors mention about DST being performed, but was it done as a part of diagnostic protocol for these patients or for the study; as majority of them were drug resistant. The major cause of treatment failure seems to be the non-identification of drug resistant cases upfront. A flow chart of the study will help clarify this, in accordance with STARD guidelines.

• What were the subjects´ characteristics i.e. HIV status? presence of diabetes? history of smoking or any other factors. This will help identify the risk factors of treatment failure and poor compliance for the purpose of predicting risk groups to improve treatment outcomes.

• The authors should mention strengths and limitations of the study, provide a proper conclusion, without overstating the results. Also, the Abstract does not really bring out the total work done. It should be rewritten.

Minor

• Please add line numbers

• Abstract, Pg 2, Para 2. 46+30, not adding up to 64, rewrite if isolates had resistance to more than 1 drugs.

• Pg. 3, Para 3. Not 512 and 528, its 511 and 526.

• Pg. 4, Para 1. inhA gene or inhA promoter region? If inhA promoter then mutations are not associated with AA change.

• Pg. 4, Para3. Guidelines for MDR-TB treatment have been updated in 2020 and the Definition of XDR-TB has been revised in 2020.

• Pg. 6, Para 2. "All culture positive MGIT 960 vials were ......performed" add reason or reference.

• Pg. 7 Para 3. Correct 0,5 to 0.5

• Pg. 8 Para1. Explain three biological replicates.

• Pg No.8 Para 2. Table S1 is list of primers.

• Pg. 9 Para 2. “Notably, serine 314 …… MDR/XDR isolates”. correct, 315

• Pg. 11 Para 2. “The mean MICs of rifampicin and isoniazid in MDR isolates were 5. 0 ug/ml and 1.5 ug /ml respectively” This detail should be moved to 'Materials and Methods' section.

• Pg. 24 Add abbreviation of ND and NA

• Pg. 24, Table footnote, change to NSM as used in table.

• Please give either Figure 1 or Table 1. The data is repeated.

• Figure 3A. What is the reason behind the low expression of Rv1258 in treatment failure resistant isolates?

6. PLOS authors have the option to publish the peer review history of their article (what does this mean?). If published, this will include your full peer review and any attached files.

Reviewer #1: **Yes: **Azar Dokht Khosravi

Reviewer #2: No

---

## [Author Response · Author response to Decision Letter 0]

15 Jul 2023

Reviewer #1: 

This study highlights the drug resistance profiles and related gene mutations in M. tuberculosis in Pakistan based on phenotypic and genotypic studies. As M. tuberculosis drug resistance including MDR and XDR phenotypes are a serious concern worldwide, it is of great importance to look at the antibiotic resistance patterns for proper treatment and patient management. Therefore, this report is a good compilation of work. However, I have some concerns as mentioned below, and upon clarifying some issues, the manuscript can be accepted for publication in PloS One.

Abstract:

-Please change the starting phrase in abstract to a correct way: "To treat tuberculosis is relatively a complicated procedure……

Done

- in the abstract results section, please add the number and % of MDR and XDR cases.

Authors´ reply: 

The abstract is now modified with additional details and, the starting phrase is changed.

Introduction:

-According to the reference 4, the first line anti-TB drugs for the treatment of non-MDR cases of TB (group 1) are Isoniazid, rifampicin, pyrazinamide and ethambutol as core drugs, streptomycin is no longer used routinely.

So why you included the latter in the anti-TB regimen? How you manage to use that as an injectable drug for duration of treatment? Please explain that in the discussion.

Authors´ reply: 

It is correct that Streptomycin is used as 2nd line drug and the patients in this study were not receiving this drug. The statement in the introduction is corrected now.

Methods:

-The information regarding the cases should well-explain at the beginning of methods section, for instance 1200 patients were confirmed TB cases, among them you found 64 drug-resistant cases and for comparison you used the same number naïve cases.

Authors´ reply: 

The Methodology is now further elaborated to make it more understandable. In addition, a supplementary figure is added to illustrate workflow adopted in the study. 

-What does (65-128) means?

Authors´ reply: 

It referred to isolate numbers in the Table. However, it is omitted from the text now to avoid any confusion. 

-In Drug susceptibility testing (DST) section, please put the reference number together as [16-18]. However why you included 3 references for a single method?

Authors´ reply: 

Corrected 

Results:

-As I noticed the rate of MDR was high in this study, again is better when you are talking about resistance to Isoniazid and Rifampicin, add the word "simultaneous" to emphasis that they are MDR. I suggest to re-write the sentence as:

"Among these sixty-four isolates, 46 isolates (71.9%) were simultaneously resistant to rifampicin and isoniazid (MDR phenotype)."

Authors´ reply: 

Done 

-In naïve group you had also 4.6% MDR (with simultaneous resistance to rifampicin and isoniazid ), you did not explain in the discussion about these, since it seems that these are the cases of primary resistance. Have you investigated the primary and secondary resistance in this research project? This should be included as the information is very useful about the circulating resistant phenotypes in a community.

Authors´ reply: 

The patients with treatment failure were primarily infected with Non-MDR M. tuberculosis as detected through Genexpert as a routine diagnosis procedure before the start of treatment plan. Apparently, they acquired resistance during treatment. Discussion is now more elaborated in this context. 

-you have bold the amikacin resistance in your study when is used for the treatment of MDR –TB cases. According to WHO the drug of choice for the treatment of first-line anti-TB treatment failure are aminoglycosides and fluoroquinolones. I did not find any statements in this section regarding the number and rate of XDR-TB, however in Table 2 it is mentioned.

What was your criteria for determining XDR strain? Since according to the WHO definition for XDR (resistance to fluoroquinolones and one of the injectable aminoglycosides), and you reported only one case with moxifloxacin.

Authors´ reply:

 Although, this study was conducted prior to revised definition of XDR-TB, we have now revised definition of XDR isolates and reformulated tables and results according to current definition of XDR M. tuberculosis. According to revised definition, all of these isolates were either MDR or non MDR. 

-For XDR cases, please add a column in Table 1 and explain more in discussion section.

Authors´ reply: 

The results are now revised considering revised definition of XDR M. tuberculosis. The XDR isolate was not found as per revised definition. The tables are now modified accordingly. 

Reviewer #2: The authors in this study entitled “Antimicrobial drug resistant features of Mycobacterium tuberculosis associated with treatment failure in Pakistan” try to elucidate antimicrobial drug resistant features of Mycobacterium tuberculosis associated with treatment failure and offer appropriateness of the erstwhile second line drugs and the efflux pump inhibitor verapamil against M. tuberculosis isolates associated with treatment failure. The manuscript targets an important issue; however, it needs to be substantially improved before consideration for publication. The authors should also perform a general proof reading for typographical errors in the manuscript. My comments are as below.

Authors´ reply: 

After an extensive proof reading of the manuscript, typographical errors are corrected now. 

• WHO has recently changed the algorithm of Drug resistant TB treatment and redefined XDR-TB. Please discuss results according to the new treatment guidelines. Even though the patients were collected in 2 sets one in 2016-17 and other recently. When were fresh samples collected? How were these patients treated? How long was the follow up done?

As mentioned by the reviewer, the study was conducted before the WHO had revised the criteria for MDR and XDR M. tuberculosis. Accordingly, in the revised version, we have now modified our discussion with respect to revised definition of MDR/XDR M. tuberculosis.

• What was the status of DST for these patients when diagnosed, when they were put on 1st line treatment. What were the results of the molecular tests for these patients, was MGIT-DST, Xpert or LPA done on these patients?

Authors´ reply: 

The treatment was started based on the result of Gene Xpert at the time of diagnosis. The isolates were non MDR at the time of diagnosis based on Gene Expert. This explanation has now further clarified in the methodology section. 

• The authors mention about DST being performed, but was it done as a part of diagnostic protocol for these patients or for the study; as majority of them were drug resistant. The major cause of treatment failure seems to be the non-identification of drug resistant cases upfront. A flow chart of the study will help clarify this, in accordance with STARD guidelines.

Authors´ reply: 

The treatment was started based on the result of Gene Expert as a part of diagnostic protocol. However, DST was not performed before the start of treatment. The DST was performed in isolates collected from the sputum of patients after treatment failure as a part of this study. Therefore, non-identification of Gene Expert in the diagnosis of MDR TB or/and acquisition of resistance during treatment can be possible cause of treatment failure. This explanation has now further elaborated in the last paragraph of discussion. The flow chart is now added as supplementary figure S1.

• What were the subjects´ characteristics i.e. HIV status? presence of diabetes? history of smoking or any other factors. This will help identify the risk factors of treatment failure and poor compliance for the purpose of predicting risk groups to improve treatment outcomes.

Authors´ reply: 

These are very important suggestions to consider. However, due to limited access to the patient’s information set by the institutional ethical review committee, this information could not be collected unfortunately. 

• The authors should mention strengths and limitations of the study, provide a proper conclusion, without overstating the results. Also, the Abstract does not really bring out the total work done. It should be rewritten.

Authors´ reply: 

Strengths and limitations are now incorporated at the end of discussion. The conclusions and abstract are revised now according to suggestions

Minor

• Please add line numbers

Authors´ reply: 

Done

• Abstract, Pg 2, Para 2. 46+30, not adding up to 64, rewrite if isolates had resistance to more than 1 drugs.

Authors´ reply: 

Done

• Pg. 3, Para 3. Not 512 and 528, its 511 and 526.

Authors´ reply: 

Corrected 

• Pg. 4, Para 1. inhA gene or inhA promoter region? If inhA promoter then mutations are not associated with AA change.

Authors´ reply: 

The mutations mentioned here are in inhA gene, not in the promoter region

• Pg. 4, Para3. Guidelines for MDR-TB treatment have been updated in 2020 and the Definition of XDR-TB has been revised in 2020.

Authors´ reply: 

Corrected according to revised definition 

• Pg. 6, Para 2. "All culture positive MGIT 960 vials were ......performed" add reason or reference.

Authors´ reply: 

The sentence is further elaborated with desired justification. 

• Pg. 7 Para 3. Correct 0,5 to 0.5

Authors´ reply: 

Corrected

• Pg. 8 Para1. Explain three biological replicates.

Authors´ reply: 

“3 biological replicates” means that, the experiment was repeated three times by growing bacteria in different vials each time. 

• Pg No.8 Para 2. Table S1 is list of primers.

Authors´ reply: 

Corrected

• Pg. 9 Para 2. “Notably, serine 314 …… MDR/XDR isolates”. correct, 315

Authors´ reply: 

Corrected

• Pg. 11 Para 2. “The mean MICs of rifampicin and isoniazid in MDR isolates were 5. 0 ug/ml and 1.5 ug /ml respectively” This detail should be moved to 'Materials and Methods' section.

Authors´ reply: 

Done 

• Pg. 24 Add abbreviation of ND and NA

Authors´ reply: 

Done 

• Pg. 24, Table footnote, change to NSM as used in table.

Authors´ reply: 

Done

• Please give either Figure 1 or Table 1. The data is repeated.

Authors´ reply: 

We consider that both figure 1 and table 1 complement each other to make results more understandable. Particularly, Circos and venn diagrams help to illustrate all combinations and frequencies of simultaneous resistance different drugs. Therefore, we prefer to keep both table 1 and figure 1. 

• Figure 3A. What is the reason behind the low expression of Rv1258 in treatment failure resistant isolates?

Authors´ reply: 

This is a very interesting question to address in the follow up studies. Since, these isolates had somehow exposed to drugs during the treatment. We reason that some response to initiate adaptive drug resistant with the involvement of Rv1258 efflux pump might have been activated in these isolates

---

## [Decision Letter · Decision Letter 1]

30 Aug 2023

PONE-D-22-27870R1Antimicrobial drug resistant features of Mycobacterium tuberculosis associated with treatment failure in PakistanPLOS ONE

Dear Dr. Ahmad,

Thank you for submitting your manuscript to PLOS ONE. After careful consideration, we feel that it has merit but does not fully meet PLOS ONE’s publication criteria as it currently stands. Therefore, we invite you to submit a revised version of the manuscript that addresses the points raised during the review process.

We look forward to receiving your revised manuscript.

Kind regards,

Atul Vashist, PhD

Academic Editor

PLOS ONE

Journal Requirements:

Reviewers' comments:

Reviewer's Responses to Questions

**Comments to the Author**

1. If the authors have adequately addressed your comments raised in a previous round of review and you feel that this manuscript is now acceptable for publication, you may indicate that here to bypass the “Comments to the Author” section, enter your conflict of interest statement in the “Confidential to Editor” section, and submit your "Accept" recommendation.

Reviewer #2: (No Response)

2. Is the manuscript technically sound, and do the data support the conclusions?

Reviewer #2: Partly

3. Has the statistical analysis been performed appropriately and rigorously? 

Reviewer #2: Yes

4. Have the authors made all data underlying the findings in their manuscript fully available?

Reviewer #2: Yes

5. Is the manuscript presented in an intelligible fashion and written in standard English?

Reviewer #2: No

6. Review Comments to the Author

Reviewer #2: The authors are advised to recheck the manuscript for typographical errors as they are still present throughout the manuscript. All standard tests should be quoted appropriately eg. Gene XPert MTB/RIF is written as Gene Expert through out; figure titles 'resistant' is spelled as ressitant . There are many spelling errors which should be corrected.

2. After revising the manuscript the Discussion portion does not seem homogenous. It should be rewritten.

7. PLOS authors have the option to publish the peer review history of their article (what does this mean?). If published, this will include your full peer review and any attached files.

Reviewer #2: No

---

## [Author Response · Author response to Decision Letter 1]

8 Sep 2023

Reviewer #2: The authors are advised to recheck the manuscript for typographical errors as they are still present throughout the manuscript. All standard tests should be quoted appropriately eg. Gene XPert MTB/RIF is written as Gene Expert through out; figure titles 'resistant' is spelled as ressitant . There are many spelling errors which should be corrected.

Authors´ reply: 

We are thankful to the reviewer for bringing the attention to typographical errors. We have now fully proofread the manuscript to remove such errors. 

2. After revising the manuscript the Discussion portion does not seem homogenous. It should be rewritten.

Authors´ reply: 

We agree with reviewer that the discussion was not homogenous. We have now rewritten the discussion to make it more coherent and homogeneous.

---

## [Decision Letter · Decision Letter 2]

8 Oct 2023

Antimicrobial drug resistant features of Mycobacterium tuberculosis associated with treatment failure

PONE-D-22-27870R2

Dear Dr. Ahmed,

We’re pleased to inform you that your manuscript has been judged scientifically suitable for publication and will be formally accepted for publication once it meets all outstanding technical requirements.

Kind regards,

Atul Vashist, PhD

Academic Editor

PLOS ONE

Additional Editor Comments (optional):

Reviewers' comments:

Reviewer's Responses to Questions

**Comments to the Author**

1. If the authors have adequately addressed your comments raised in a previous round of review and you feel that this manuscript is now acceptable for publication, you may indicate that here to bypass the “Comments to the Author” section, enter your conflict of interest statement in the “Confidential to Editor” section, and submit your "Accept" recommendation.

Reviewer #2: All comments have been addressed

2. Is the manuscript technically sound, and do the data support the conclusions?

Reviewer #2: Yes

3. Has the statistical analysis been performed appropriately and rigorously? 

Reviewer #2: N/A

4. Have the authors made all data underlying the findings in their manuscript fully available?

Reviewer #2: Yes

5. Is the manuscript presented in an intelligible fashion and written in standard English?

Reviewer #2: Yes

6. Review Comments to the Author

Reviewer #2: (No Response)

7. PLOS authors have the option to publish the peer review history of their article (what does this mean?). If published, this will include your full peer review and any attached files.

Reviewer #2: No

---

## [Editor Report · Acceptance letter]

17 Oct 2023

PONE-D-22-27870R2 

Antimicrobial drug resistant features of *Mycobacterium tuberculosis* associated with treatment failure 

Dear Dr. Ahmad:

I'm pleased to inform you that your manuscript has been deemed suitable for publication in PLOS ONE. Congratulations! Your manuscript is now with our production department. 

Kind regards, 

on behalf of

Dr. Atul Vashist 

Academic Editor

PLOS ONE